# Understanding and Exploiting the Low-Rank Structure of Deep Networks

## Abstract

Training methods for deep networks are primarily variants on stochastic gradient descent. Techniques that use (approximate) second-order information are rarely used because of the computational cost and noise associated with those approaches in deep learning contexts. However, in this paper, we show how feedforward deep networks exhibit a low-rank derivative structure. This low-rank structure makes it possible to use second-order information without needing approximations and without incurring a significantly greater computational cost than gradient descent. To demonstrate this capability, we implement Cubic Regularization (CR) on a feedforward deep network with stochastic gradient descent and two of its variants. There, we use CR to calculate learning rates on a per-iteration basis while training on the MNIST and CIFAR-10 datasets. CR proved particularly successful in escaping plateau regions of the objective function. We also found that this approach requires less problem-specific information (e.g. an optimal initial learning rate) than other first-order methods in order to perform well.

## 1 Introduction

### 1.1 Gradient-Based Optimization and Deep Learning

Gradient-based optimization methods use derivative information to determine intelligent search directions when minimizing a continuous objective function. The steepest descent method is the most basic of these optimization techniques, but it is known to converge very slowly in ill-conditioned systems. Even outside of these cases, it still only has a linear rate of convergence. Newton's method is a more sophisticated approach – one that uses second-order derivative information, which allows the optimizer to model the error surface more accurately and thus take more efficient update steps. When it converges, it does so quadratically, but Newton's method also has limitations of its own. Firstly, it does not scale well: it can be very expensive to calculate, store, and invert the objective function Hessian. Secondly, the method may fail if the Hessian is indefinite or singular.

A variety of methods have been developed to try and appropriate the strengths of each approach while avoiding their weaknesses. The conjugate gradient method, for example, uses only first-order information but uses the history of past steps taken to produce a better convergence rate than steepest descent. Quasi-Newton methods, on the other hand, approximate the Hessian (or its inverse) using first-order information and may enforce positive-definiteness on its approximation. Other approaches like trust region methods use second-order information without requiring convexity. For further information about gradient-based optimization, see Nocedal & Wright (2006).

Deep learning (DL) provides a set of problems that can be tackled with gradient-based optimization methods, but it has a number of unique features and challenges. Firstly, DL problems can be extremely large, and storing the Hessian, or even a full matrix approximation thereto, is not feasible for such problems. Secondly, DL problems are often highly nonconvex. Thirdly, training deep networks via mini-batch sampling results in a stochastic optimization problem. Even if the necessary expectations can be calculated (in an unbiased way), the variance associated with the batch sample calculations produces noise, and this noise can make it more difficult to perform the optimization. Finally, deep networks consist of the composition of analytic functions whose forms are known. As such, we can calculate derivative information analytically via back-propagation (i.e. the chain rule).

## 1.2 Training Methods for Deep Learning

These special characteristics of DL have motivated researchers to develop training methods specifically designed to overcome the challenges with training a deep neural network. One such approach is layer-wise pretraining (Bengio et al., 2007), where pretraining a neural network layer-by-layer encourages the weights to initialize close to a optimal minimum. Transfer learning (Yosinski et al., 2014) works by a similar mechanism, relying on knowledge gained through previous tasks to encourage nice training on a novel task. Outside of pretraining, a class of optimization algorithms have been specifically designed for training deep networks. The Adam, Adagrad, and Adamax set of algorithms provide examples of using history-dependent learning rate adjustment (Kingma & Ba, 2014). Similarly, Nesterov momentum provides a method for leveraging history dependence in stochastic gradient descent (Sutskever et al., 2013). One could possibly argue that these methods implicitly leverage second order information via their history dependence, but the stochastic nature of mini-batching prevents this from becoming explicit.

Some researchers have sought to use second-order information explicitly to improve the training process. Most of these methods have used an approximation to the Hessian. For example, the L-BFGS method can estimate the Hessian (or its inverse) in a way that is feasible with respect to memory requirements; however, the noise associated with the sampling techniques can either overwhelm the estimation or require special modifications to the L-BFGS method to prevent it from diverging (Byrd et al., 2016). There have been two primary ways to deal with this: subsampling (Byrd et al., 2016; Moritz et al., 2016) and mini-batch reuse (Schraudolph et al., 2007; Mokhtari & Ribeiro, 2014). Subsampling involves updating the Hessian approximation every $L$ iterations rather than every iteration, as would normally be done. Mini-batch reuse consists of using the same mini-batch on subsequent iterations when calculating the difference in gradients between those two iterations. These approximate second-order methods typically have a computational cost that is higher than, though on the same order of, gradient descent, and that cost can be further reduced by using a smaller mini-batch for the Hessian approximation calculations than for the gradient calculation (Byrd et al., 2011). There is also the question of bias: it is possible to produce unbiased low-rank Hessian approximations (Martens et al., 2012), but if the Hessian is indefinite, then quasi-Newton methods will prefer biased estimates – ones that are positive definite. Other work has foregone these kinds of Hessian approximations in favor of using finite differences (Martens, 2010).

## 1.3 Contributions

In this paper, we prove, by construction, that the first and second derivatives of feedforward deep learning networks exhibit a low-rank, outer product structure. This structure allows us to use and manipulate second-order derivative information, without requiring approximation, in a computationally feasible way. As an application of this low-rank structure, we implement Cubic Regularization (CR) to exploit Hessian information in calculating learning rates while training a feedforward deep network. Finally, we show that calculating learning rates in this fashion can improve existing training methods' ability to exit plateau regions during the training process.

## 2 The Low-Rank Structure of Deep Network Derivatives

Second-order derivatives are not widely used in DL, and where they *are* used, they are typically estimated. These derivatives can be calculated analytically, but this is not often done because of the scalability constraints described in Section 1.1. If we write out the first and second derivatives, though, we can see that they have a low-rank structure to them – an outer product structure, in fact. When a matrix has low rank (or less than full rank), it means that the information contained in that matrix (or the operations performed by that matrix) can be fully represented without needing to know every entry of that matrix. An outer product structure is a special case of this, where an $mxn$ matrix $\mathbf{A}$ can be fully represented by two vectors $\mathbf{A} = \mathbf{u}\mathbf{v}^T$. We can then calculate, store, and use second-order derivatives exactly in an efficient manner by only dealing with the components needed to represent the full Hessians rather than dealing with those Hessians themselves. Doing this involves some extra calculations, but the storage costs are comparable to those of gradient calculations.

In this section, we will illustrate the low-rank structure for a feedforward network, of arbitrary depth and layer widths, consisting of ReLUs in the hidden layers and a softmax at the output layer. A

feedforward network with arbitrary activation functions has somewhat more complicated derivative formulae, but those derivatives still exhibit a low-rank structure. That structure also does not depend on the form of the objective function or whether a softmax is used, and it is present for convolutional and recurrent layers as well. The complete derivations for these cases are given in Appendix B.

In our calculations, we make extensive use of index notation with the summation convention (Ivancevic & Ivancevic, 2007). In index notation, a scalar has no indices ($v$), a vector has one index ($\mathbf{v}$ as $v^i$ or $v_i$), a matrix has two ($\mathbf{V}$ as $V^{ij}$, $V^i_j$, or $V_{ij}$), and so on. The summation convention holds that repeated indices in a given expression are summed over unless otherwise indicated. For example, $\mathbf{a}^T\mathbf{b} = \sum_i a^i b_i = a^i b_i$. The pair of indices being summed over will often consist of a superscript and a subscript; this is a bookkeeping technique used in differential geometry, but in this context, the subscripting or superscripting of indices will not indicate covariance or contravariance. We have also adapted index notation slightly to suit the structure of deep networks better: indices placed in brackets (e.g. the $k$ in $v^{(k),j}$) are not summed over, even if repeated, unless explicitly indicated by a summation sign. A tensor convention that we *will* use, however, is the Kronecker delta: $\delta^{ij}$, $\delta^i_j$, or $\delta_{ij}$. The Kronecker delta is the identity matrix represented in index notation: it is 1 for $i = j$ and 0 otherwise. The summation convention can sometimes be employed to simplify expressions containing Kronecker deltas. For example, $\delta^j_i v^i = v^j$ and $\delta_{ij} V_{jk} = V_{ik}$.

Let us consider a generic feedforward network with ReLU activation functions in $n$ hidden layers, a softmax at the output layer, and categorical cross-entropy as the objective function (defined in more detail in Appendix B. The first derivatives, on a per-sample basis, for this deep network are

$$\frac{\partial f}{\partial u^i_j} = \frac{\partial f}{\partial p^i} v^{(n),j} \tag{1}$$

$$\frac{\partial f}{\partial w^{(k),i}_j} = \frac{\partial f}{\partial p^l} u^l_m \eta^{(n,k),m}_i v^{(k-1),j} \tag{2}$$

where $f$ is the per-sample objective function, $v^{(k),j}$ is the vector output of layer $k$, $u^i_j$ is the matrix of weights in the softmax, and $p^j = u^j_i v^{(n),i}$ (which is the vector quantity evaluated by the softmax). For the full derivation, and the definition of the matrix quantity $\eta^{(n,k),m}_i$, see Appendix B. In calculating these expressions, we have deliberately left $\frac{\partial f}{\partial p^j}$ unevaluated. This keeps the expression relatively simple, and programs like TensorFlow (Abadi et al., 2015) can easily calculate this for us. Leaving it in this form also preserves the generality of the expression – there is no low-rank structure contained in $\frac{\partial f}{\partial p^j}$, and the low-rank structure of the network as a whole is therefore shown to be independent of the objective function and whether or not a softmax is used. In fact, as long as Equation 13 holds, any sufficiently smooth function of $p^j$ may be used in place of a softmax without disrupting the low-rank structure. The one quantity that needs to be stored here is $\eta^{(n,k),j}_i$ for $k = 1, 2, \ldots, n - 1$; it will be needed in the second derivative calculations. Note, however, that this is roughly the same size as the gradient itself.

We can now see the low-rank structure: $\frac{\partial f}{\partial u^i_j}$ is the outer product (or tensor product) of the vectors $\frac{\partial f}{\partial p^i}$ and $v^{(n),j}$, and $\frac{\partial f}{\partial w^{(k),i}_j}$ is the outer product of $\frac{\partial f}{\partial p^l} u^l_m \eta^{(n,k),m}_i$ (which ends up being a rank-1 tensor) and $v^{(k-1),j}$. The index notation makes the outer product structure clear. It is important to note that this low-rank structure only exists *for each sample* – a weighted sum of low-rank matrices is not necessarily (and generally, will not be) low rank. In other words, even if the gradient of $f$ is low rank, the gradient of the expectation, $F = E[f]$, will not be, because the gradient of $F$ is the weighted sum of the gradients of $f$. The second-order objective function derivatives are then

$$\frac{\partial^2 f}{\partial u^i_j \partial u^s_t} = \frac{\partial^2 f}{\partial p^i \partial p^s} v^{(n),j} v^{(n),t} \tag{3}$$

$$\frac{\partial^2 f}{\partial u^i_j \partial w^{(k),s}_t} = \frac{\partial f}{\partial p^i} \eta^{(n,k),j}_s v^{(k-1),t} + \frac{\partial^2 f}{\partial p^i \partial p^l} v^{(n),j} u^l_m \eta^{(n,k),m}_s v^{(k-1),t} \tag{4}$$

$$\frac{\partial^2 f}{\partial w_j^{(k),i} \partial w_t^{(r),s}} = \frac{\partial^2 f}{\partial p^l \partial p^q} u_m^l \eta_i^{(n,k),m} v^{(k-1),j} u_a^q \eta_s^{(n,r),a} v^{(r-1),t}$$

$$+ \frac{\partial f}{\partial p^l} u_m^l \times \begin{cases} \eta_s^{(n,r),m} \eta_i^{(r-1,k),t} v^{(k-1),j} & r > k \\ 0 & r = k \\ \eta_i^{(n,k),m} \eta_s^{(k-1,r),j} v^{(r-1),t} & r < k \end{cases} \tag{5}$$

Calculating all of these second derivatives requires the repeated use of $\frac{\partial^2 f}{\partial \mathbf{p}^2}$. Evaluating that Hessian is straightforward given knowledge of the activation functions and objective used in the network, and storing it is also likely not an issue as long as the number of categories is small relative to the number of weights. For example, consider a small network with 10 categories and 1000 weights. In such a case, $\frac{\partial^2 f}{\partial \mathbf{p}^2}$ would only contain 100 entries – the gradient would be 10 times larger. We now find that we have to store $\eta_j^{(n,k),i}$ values in order to calculate the derivatives. In $\frac{\partial^2 f}{\partial \mathbf{w}^2}$, we also end up needing $\eta_j^{(r,k),i}$ for $r \neq n$. In a network with $n$ hidden layers, we would then have $\frac{n(n-1)}{2}$ of the $\eta_j^{(r,k),i}$ matrices to store. For $n = 10$, this would be 45, for $n = 20$, this would be 190, and so on. This aspect of the calculations does not seem to scale well, but in practice, it is relatively simple to work around. It is still necessary to store $\eta_j^{(n,k),i}$, $k < n$, but $\eta_j^{(r,k),i}$, $r < n$, only actually shows up in one place, and thus it is possible to calculate each $\eta_j^{(r,k),i}$ matrix, use it, and discard it without needing to store it for future calculations. The key thing to note about these second derivatives is that they retain a low-rank structure – they are now tensor products (or the sums of tensor products) of matrices and vectors. For example,

$$\frac{\partial^2 f}{\partial u_j^i \partial w_t^{(k),s}} = \left( \frac{\partial f}{\partial p^i} \times \eta_s^{(n,k),j} \times v^{(k-1),t} \right) + \left( a_{is} \times v^{(n),j} \times v^{(k-1),t} \right) \tag{6}$$

$$a_{is} = \frac{\partial^2 f}{\partial p^i \partial p^l} u_m^l \eta_s^{(n,k),m} \tag{7}$$

With these expressions, it would be relatively straightforward to extract the diagonal of the Hessian and store or manipulate it as a vector. The rank of the weighted sum of low rank components (as occurs with mini-batch sampling) is generally larger than the rank of the summed components, however. As such, manipulating the entire Hessian may not be as computationally feasible; this will depend on how large the mini-batch size is relative to the number of weights. The low rank properties that we highlight here for the Hessian exist on a per-sample basis, as they did for the gradient, and therefore, the computational savings provided by this approach will be most salient when calculating scalar or vector quantities on a sample-by-sample basis and then taking a weighted sum of the results. In principle, we could calculate third derivatives, but the formulae would likely become unwieldy, and they may require memory usage significantly greater than that involved in storing gradient information. Second derivatives should suffice for now, but of course if a use arose for third derivatives, calculating them would be a real option. Thus far, we have not included bias terms. Including bias terms as trainable weights would increase the overall size of the gradient (by adding additional variables), but it would not change the overall low-rank structure. Using the calculations provided in Appendix B, it would not be difficult to produce the appropriate derivations.

## 3 CUBIC REGULARIZATION IN DEEP LEARNING

Cubic Regularization (CR) is a trust region method that uses a cubic model of the objective function:

$$f(\mathbf{x}) \approx m_j(\mathbf{s}_j) = f(\mathbf{x}_j) + \mathbf{s}_j^T \frac{\partial f}{\partial \mathbf{x}} + \frac{1}{2} \mathbf{s}_j^T \mathbf{H}_j \mathbf{s}_j + \frac{1}{6} \sigma_j \|\mathbf{s}_j\|^3 \tag{8}$$

at the $j$-th iteration, where $\mathbf{H}_j$ is the objective function Hessian and $\mathbf{s}_j = \mathbf{x} - \mathbf{x}_j$. The cubic term makes it possible to use information in the Hessian without requiring convexity, and the weight $\sigma_j$ on

that cubic term can have its own update scheme (based on how well $m(\mathbf{s}_j)$ approximates $f$ (Kohler & Lucchi, 2017)). Solving for an optimal $\mathbf{s}_j$ value then involves finding the root of a univariate nonlinear equation (Nesterov & Polyak, 2006). CR is not commonly used in deep learning; we have seen only one example of CR applied to machine learning (Kohler & Lucchi, 2017) and no examples with deep learning. This is likely the case because of two computationally expensive operations: calculating the Hessian and solving for $\mathbf{s}_j$. We can overcome the first by using the low-rank properties described above. The second is more challenging, but we can bypass it by using CR to calculate a step length (i.e. the learning rate) for a given search direction rather than calculating the search direction itself.

## 3.1 IMPLEMENTATION

Our approach in this paper is to use CR as a metamethod – a technique that sits on top of existing training algorithms. The algorithm calculates a search direction, and then CR calculates a learning rate for that search direction. For a general iterative optimization process, this would look like $\mathbf{x}_{j+1} = \mathbf{x}_j + \alpha_j \mathbf{g}_j$, where $\mathbf{g}_j$ is the search direction (which need not be normalized), $\alpha_j$ is the learning rate, and the subscript refers to the iteration. With the search direction fixed, $m$ would then be a cubic function of $\alpha$ at each iteration. Solving $\frac{\partial m}{\partial \alpha} = 0$ as a quadratic equation in $\alpha$ then yields

$$\alpha = \frac{-\mathbf{g}^T \mathbf{H} \mathbf{g} \pm \sqrt{\left(\mathbf{g}^T \mathbf{H} \mathbf{g}\right)^2 - 2\left(\mathbf{g}^T \nabla f\right)\left(\sigma \|\mathbf{g}\|^3\right)}}{\sigma \|\mathbf{g}\|^3} \tag{9}$$

If we assume that $\mathbf{g}^T \nabla f < 0$ (i.e. $\mathbf{g}$ is a descent direction), then $\alpha$ is guaranteed to be real. Continuing under that assumption, of the two possible $\alpha$ values, we choose the one guaranteed to be positive. The sampling involved in mini-batch training means that there are a number of possible ways to get a final $\alpha_j \mathbf{g}_j$ result. One option would be to calculate $E[\alpha_j \mathbf{g}_j]$. This would involve calculating an $\alpha$ value with respect to the search direction produced by each sample point and then averaging the product $\alpha \mathbf{g}$ over all of the sample points. Doing this should produce an unbiased estimate of $\alpha_j \mathbf{g}_j$, but in practice, we found that this approach resulted in a great deal sampling noise and thus was not effective. The second approach would calculate $E[\alpha_j] \times E[\mathbf{g}_j]$. To do this, we would calculate an $\alpha$ value with respect to the search direction produced by each sample point, as in the first option, calculate an average $\alpha$ value, and multiply the overall search direction by that average. This approach, too, suffered from excessive noise. In the interest of reducing noise and increasing simplicity, we chose a third option: once the step direction had been determined, we considered that fixed, took the average of $\mathbf{g}^T \mathbf{H} \mathbf{g}$ and $\mathbf{g}^T \nabla f$ over all of the sample points to produce $m(\alpha)$ and then solved for a single $\alpha_j$ value. This approach was the most effective of the three.

## 3.2 COMPUTATIONAL RESULTS

To test CR computationally, we created deep feedforward networks using ReLU activations in the hidden layers, softmax in the output layer, and categorical cross-entropy as the error function; we then trained them on the MNIST (LeCun et al., 1998) and CIFAR-10 (Krizhevsky & Hinton, 2009) data sets. This paper shows results from networks with 12 hidden layers, each 128 nodes wide. For the purposes of this paper, we treat network training strictly as an optimization process, and thus we are not interested in network performance measures such as accuracy and validation error – the sole consideration is minimizing the error function presented to the network. As we consider that minimization progress, we will also focus on optimization iteration rather than wall clock time: the former indicates the behaviour of the algorithm itself, whereas the latter is strongly dependent upon implementation (which we do not want to address at this juncture). Overall computational cost per iteration matters, and we will discuss it, but it will not be our primary interest. Further implementation details are found in Appendix A.

Figure 1 shows an example of CR (applied on top of SGD). In this case, using CR provided little to no benefit. The average learning rate with CR was around 0.05 (a moving average with a period of 100 is shown in green on the learning rate plot both here and in the rest of the paper), which was close to our initial choice of learning rate. This suggests that 0.02 was a good choice of learning rate. Another reason the results were similar, though, is that the optimization process did not run into any

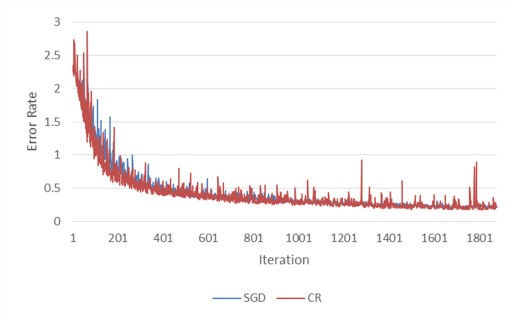

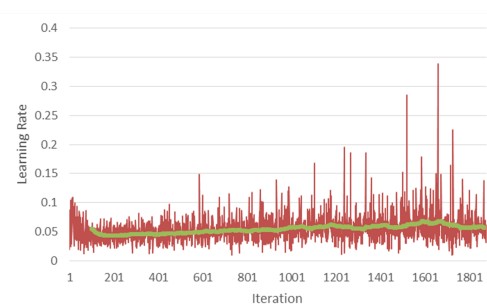

(a) Error (SGD in blue, SGD with CR in red)

(b) Learning rate (calculated rate in red, period-100 moving average in green)

Figure 1: Cubic Regularization (CR) applied to Stochastic Gradient Descent (SGD); initial learning rate = 0.01, $\sigma = 100$

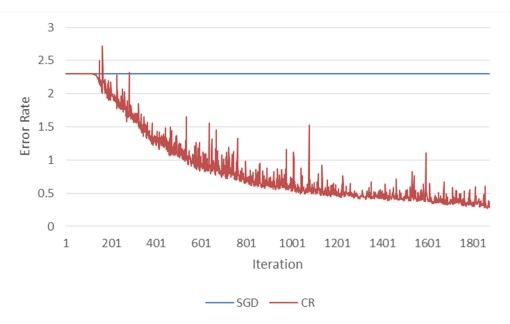

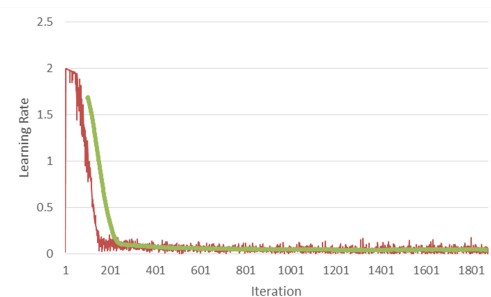

(a) Error (SGD in blue, SGD with CR in red)

(b) Learning rate (calculated rate in red, period-100 moving average in green)

Figure 2: Cubic Regularization (CR) applied to Stochastic Gradient Descent (SGD); initial learning rate = 0.02, $\sigma = 100$

plateaus. We would expect CR to provide the greatest benefit when the optimization gets stuck on a plateau – having information about the objective function curvature would enable the algorithm to increase the learning rate while on the plateau and then return it to a more typical value once it leaves the plateau. To test this, we deliberately initialized our weights so that they lay on a plateau: the objective function is very flat near the origin, and we found that setting the network weights to random values uniformly sampled between 0.1 and -0.1 was sufficient.

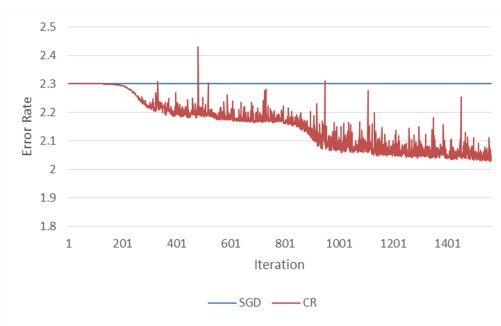

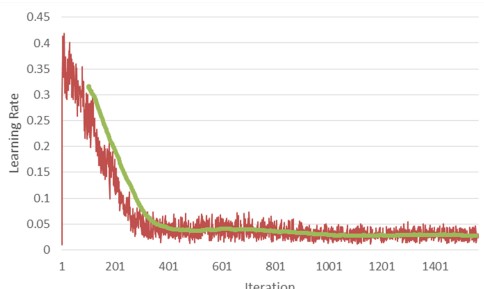

(a) Error (SGD in blue, SGD with CR in red)

(b) Learning rate (calculated rate in red, period-100 moving average in green)

Figure 3: Cubic Regularization (CR) applied to Stochastic Gradient Descent (SGD) on the CIFAR-10 Dataset; initial learning rate = 0.01, $\sigma = 1000$

Figure 2 shows the results of SGD with and without CR when stuck on a plateau. There, we see a hundred-fold increase in the learning rate while the optimization is on the plateau, but this rate drops rapidly as the optimization exits the plateau, and once it returns to a more normal descent, the learning rate also returns to an average of about 0.05 as before. The CR calculation enables the training process to recognize the flat space and take significantly larger steps as a result. Applying CR to SGD when training on CIFAR-10 (Figure 3) produced results similar to those seen on MNIST.

We then considered if this behaviour would hold true on other training algorithms: we employed CR with Adagrad (Duchi et al., 2011) and Adadelta (Zeiler, 2012)on MNIST. The results were similar. CR did not provide a meaningful difference when the algorithms performed well, but when those algorithms were stuck on plateaus, CR increased the learning rate and caused the algorithms to exit the plateau more quickly than they otherwise would have (Figures 4 and 5). The relative magnitudes of those increases were smaller than for SGD, but Adagrad and Adadelta already incorporate some adaptive learning rate behaviour, and good choices for the initial learning rate varied significantly from algorithm to algorithm. We also used a larger value for $\sigma$ to account for the increased variability due to those algorithms' adaptive nature. The result with Adadelta showed some interesting learning rate changes: the learning rate calculated by CR dropped steadily as the algorithm exited the plateau, but it jumped again around iteration 1200 as it apparently found itself in a flat region of space.

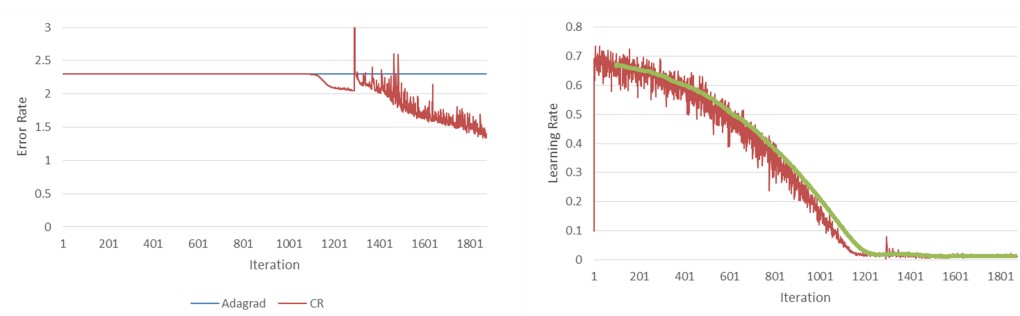

(a) Error (Adagrad in blue, Adagrad with CR in red)  (b) Learning rate (calculated rate in red, period-100 moving average in green)

Figure 4: Cubic Regularization (CR) applied to Adagrad; initial learning rate = 0.1, $\sigma = 1000$

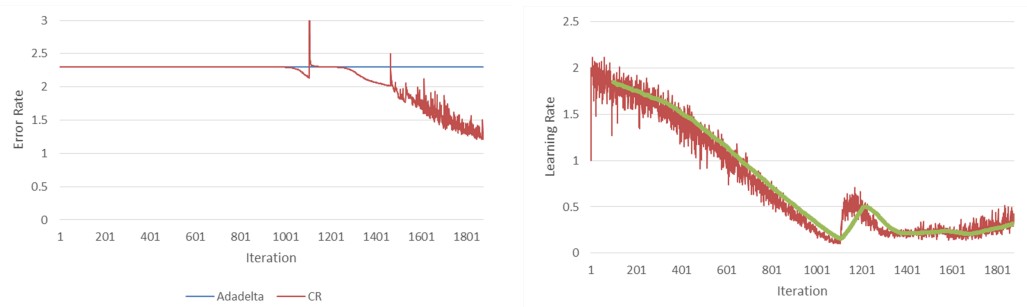

(a) Error (Adadelta in blue, Adadelta with CR in red)  (b) Learning rate (calculated rate in red, period-100 moving average in green)

Figure 5: Cubic Regularization (CR) applied to Adadelta; initial learning rate = 1.0, $\sigma = 1000$

## 4 DISCUSSION

We see this CR approach as an addition to, not a replacement for, existing training methods. It could potentially replace existing methods, but it does not have to in order to be used. Because of the low-rank structure of the Hessian, we can use CR to supplement existing optimizers that do not explicitly leverage second order information. The CR technique used here is most useful when the optimization is stuck on a plateau prior to convergence: CR makes it possible to determine whether

the optimization has converged (perhaps to a local minimum) or is simply bogged down in a flat region. It may eventually be possible to calculate a search direction as well as a step length, which would likely be a significant advancement, but this would be a completely separate algorithm.

We found that applying CR to Adagrad and Adadelta provided the same kinds of improvements that applying CR to SGD did. However, using CR with Adam (Kingma & Ba, 2014) did not provide gains as it did with the other methods. Adam generally demonstrates a greater degree of adaptivity than Adagrad or Adadelta; in our experiments, we found that Adam was better than Adagrad or Adadelta in escaping the plateau region. We suspect that trying to overlay an additional calculated learning rate on top of the variable-specific learning rate produced by Adam may create interference in both sets of learning rate calculations. Analyzing each algorithm's update scheme in conjunction with the CR calculations could provide insight into the nature and extent of this interference, and provide ways to further improve both algorithms. In future work, though, it would not be difficult to adapt the CR approach to calculate layer- or variable-specific learning rates, and doing that could address this problem. Calculating a variable-specific learning rate would essentially involve rescaling each variable's step by the corresponding diagonal entry in the Hessian; calculating a layer-specific learning rate would involve rescaling the step of each variable in that layer by some measure of the block diagonal component of the Hessian corresponding to those variables. The calculations for variable-specific learning rates with CR are given in Appendix B.

There are two aspects of the computational cost to consider in evaluating the use of CR. The first aspect is storage cost. In this regard, the second-order calculations are relatively inexpensive (comparable to storing gradient information). The second aspect is the number of operations, and the second-order calculations circumvent the storage issue by increasing the number of operations. The number of matrix multiplications involved in calculating the components of Equation 9, for example, scales quadratically with the number of layers (see the derivations in Appendix B). Although the number of matrix multiplications will not change with an increase in width, the cost of naïve matrix multiplication scales cubically with matrix size. That being said, these calculations are parallelizable and as such, the effect of the computation cost will be implementation-dependent.

A significant distinction between CR and methods like SGD has to do with the degree of knowledge about the problem required prior to optimization. SGD requires an initial learning rate and (usually) a learning rate decay scheme; an optimal value for the former can be very problem-dependent and may be different for other algorithms when applied to the same problem. For CR, it is necessary to specify $\sigma$, but optimization performance is relatively insensitive to this – order of magnitude estimates seem to be sufficient – and varying $\sigma$ has a stronger affect on the *variability* of the learning rate than it does on the magnitude (though it does affect both). If the space is very curved, the choice of $\sigma$ matters little because the step size determination is dominated by the curvature, and if the space if flat, it bounds the step length. It is also possible to employ an adaptive approach for updating $\sigma$ (Kohler & Lucchi, 2017), but we did not pursue that here. Essentially, using CR is roughly equivalent to using the optimal learning rate (for SGD).

## 5   CONCLUSIONS

In this paper, we showed that feedforward networks exhibit a low-rank derivative structure. We demonstrate that this structure provides a way to represent the Hessian efficiently; we can exploit this structure to obtain higher-order derivative information at relatively low computational cost and without massive storage requirements. We then used second-order derivative information to implement CR in calculating a learning rate when supplied with a search direction. The CR method has a higher per-iteration cost than SGD, for example, but it is also highly parallelizable. When SGD converged well, CR showed comparable optimization performance (on a per-iteration basis), but the adaptive learning rate that CR provided proved to be capable of driving the optimization away from plateaus that SGD would stagnate on. The results were similar with Adagrad and Adadelta, though not with Adam. CR also required less problem-specific knowledge (such as an optimal initial learning rate) to perform well. At this point, we see it as a valuable technique that can be incorporated into existing methods, but there is room for further work on exploiting the low-rank derivative structure to enable CR to calculate search directions as well as step sizes.

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

## A  EXPERIMENTAL PROCEDURE

Starting at a point far from the origin resulted in extremely large derivative and curvature values (not to mention extremely large objective function values), and this could sometimes cause difficulties for the CR method. This was easy to solve by choosing an initialization point relatively near the origin; choosing an initialization relatively near the origin also provided a significantly better initial objective function value. We initialized the networks' weights to random values between an upper and lower bound: to induce plateau effects, we set, the bounds to $\pm 0.1$, otherwise, we set them to $\pm 0.2$.

All of the networks used a mini-batch size of 32 and were implemented in TensorFlow (Abadi et al., 2015). The initial learning rate varied with network size; we chose learning rates that were large and reasonable but perhaps not optimal, and for optimization algorithms with other parameters governing the optimization, we used the default TensorFlow values for those parameters. For the learning rate decay, we used an exponential decay with a decay rate of 0.95 per 100 iterations. The $\sigma$ value used is specified along with the initial learning rate for each network's results. This value was also not optimized but was instead set to a reasonable power of 10.

## B  LOW-RANK DERIVATIONS FOR DEEP NETWORKS

### B.1  FEEDFORWARD NETWORK WITH RELU ACTIVATIONS

Table 1 provides a nomenclature for our deep network definition.

Table 1: Nomenclature – Formulation

| Quantity | Description |
|---|---|
| $n$ | Number of hidden layers |
| $x^i$ | Vector of inputs for a single sample |
| $v^{(k),j}$ | Vector output of layer $k$ |
| $w_i^{(k),j}$ | Matrix of weights for layer $k$ |
| $\mathcal{A}(\cdot)$ | Activation function |
| $u_i^j$ | Matrix of output layer weights |
| $p^j$ | Vector of intermediate variables for the output layer |
| $\hat{y}^l$ | Vector of outputs for a single sample |
| $y^l$ | Vector of labels for a single sample |
| $f$ | Scalar objective function value for a single sample |
| $F$ | Scalar objective function |

Equations 10-16 define a generic feedforward network with ReLU activation functions in the hidden layers, $n$ hidden layers, a softmax at the output layer, and categorical cross-entropy as the objective function.

$$v^{(k),j} = \mathcal{A}\left(w_i^{(k),j}v^{(k-1),i}\right), \ k = 1, \ldots, n \tag{10}$$

$$\mathcal{A}(z) = \max(z, 0) \tag{11}$$

$$v^{(0),i} = x^i \tag{12}$$

$$p^j = u_i^j v^{(n),i} \tag{13}$$

$$\hat{y}^j = \frac{\exp\left(p^j\right)}{\sum_l \exp\left(p^l\right)} \tag{14}$$

$$f = -y^l \ln \hat{y}^l \tag{15}$$

$$F = E[f] \tag{16}$$

The relevant first derivatives for this deep network are

$$\mathcal{A}'(z) = \begin{cases} 1 & z > 0 \\ 0 & z < 0 \end{cases} \tag{17}$$

$$\frac{\partial v^{(k),j}}{\partial w_t^{(l),s}} = \begin{cases} 0 & l > k \\ \delta_s^j \mathcal{A}'\left(w_i^{(k),j}v^{(k-1),i}\right)v^{(k-1),t} & l = k \\ \mathcal{A}'\left(w_i^{(k),j}v^{(k-1),i}\right)w_q^{(k),j}\frac{\partial v^{(k-1),q}}{\partial w_t^{(l),s}} & l < k \end{cases} \tag{18}$$

where there is no summation over $j$ in Equation 18. We now define several intermediate quantities to simplify the derivation process:

$$\gamma_s^{(k),j} \equiv \delta_s^j \mathcal{A}'\left(w_i^{(k),j}v^{(k-1),i}\right) \tag{19}$$

$$\beta_i^{(k),j} \equiv \mathcal{A}'\left(w_l^{(k),j}v^{(k-1),l}\right)w_i^{(k),j} = \gamma_s^{(k),j}w_i^{(k),s} \tag{20}$$

$$\alpha_{j_l}^{(k,l),j_k} \equiv \begin{cases} \prod_{i=l+1}^k \beta_{j_{i-1}}^{(i),j_i} & k > l \\ \delta_j^i & k = l \\ 0 & k < l \end{cases} \tag{21}$$

$$\alpha_{j_l}^{(k,l),j_k}\alpha_{j_m}^{(l,m),j_l} = \alpha_{j_m}^{(k,m),j_k} \tag{22}$$

$$\eta_i^{(k,l),j} \equiv \alpha_s^{(k,l),j}\gamma_i^{(k),s} \tag{23}$$

where there is no summation over $j$ in Equations 19 and 20. We can now complete our calculations of the first derivatives.

$$\frac{\partial v^{(k),j}}{\partial w_t^{(l),s}} = \begin{cases} 0 & l > k \\ \eta_s^{(k,l),j}v^{(l-1),t} & l \leq k \end{cases} \tag{24}$$

$$\frac{\partial p^j}{\partial u_k^l} = \delta_l^j v^{(n),k} \tag{25}$$

$$\frac{\partial p^j}{\partial v^{(n),i}} = u_i^j \tag{26}$$

$$\frac{\partial f}{\partial u_j^i} = \frac{\partial f}{\partial p^k}\frac{\partial p^k}{\partial u_j^i} = \frac{\partial f}{\partial p^i}v^{(n),j} \tag{27}$$

$$\frac{\partial f}{\partial w_j^{(k),i}} = \frac{\partial f}{\partial p^l}\frac{\partial p^l}{\partial v^{(n),m}}\frac{\partial v^{(n),m}}{\partial w_j^{(k),i}} = \frac{\partial f}{\partial p^l}u_m^l \eta_i^{(n,k),m}v^{(k-1),j} \tag{28}$$

We then start our second derivative calculations by considering some intermediate quantities:

$$\mathcal{A}''(z) = 0 \tag{29}$$

$$\frac{\partial \alpha_q^{(n,k),m}}{\partial w_t^{(r),s}} = \alpha_a^{(n,r),m} \mathcal{A}'\left(w_p^{(r),a} v^{(r-1),p}\right) \delta_s^a \delta_b^t \alpha_q^{(r-1,k),b} = \eta_s^{(n,r),m} \alpha_q^{(r-1,k),t} \tag{30}$$

$$\frac{\partial \gamma_s^{(k),j}}{\partial (\cdot)} = 0 \tag{31}$$

$$\frac{\partial \eta_i^{(n,k),m}}{\partial w_t^{(r),s}} = \frac{\partial \alpha_q^{(n,k),m}}{\partial w_t^{(r),s}} \gamma_i^{(k),q} \tag{32}$$

$$\frac{\partial^2 v^{(n),m}}{\partial w_j^{(k),i} \partial w_t^{(r),s}} = \begin{cases} \frac{\partial \alpha_q^{(n,k),m}}{\partial w_t^{(r),s}} \gamma_i^{(k),q} v^{(k-1),j} & r > k \\ 0 & r = k \\ \eta_i^{(n,k),m} \frac{\partial v^{(k-1),j}}{\partial w_t^{(l),s}} & r < k \end{cases}$$

$$= \begin{cases} \eta_s^{(n,r),m} \eta_i^{(r-1,k),t} v^{(k-1),j} & r > k \\ 0 & r = k \\ \eta_i^{(n,k),m} \eta_s^{(k-1,r),j} v^{(r-1),t} & r < k \end{cases} \tag{33}$$

The second derivative of the ReLU vanishes, which simplifies the second derivative calculations significantly. Technically, the second derivative is undefined at the origin, but the singularity is removable, and thus we can define the second derivative to be 0 at the origin. We can then calculate the second-order objective function derivatives:

$$\frac{\partial^2 f}{\partial u_j^i \partial u_t^s} = \frac{\partial^2 f}{\partial p^k \partial p^l} \frac{\partial p^k}{\partial u_j^i} \frac{\partial p^l}{\partial u_t^s} = \frac{\partial^2 f}{\partial p^i \partial p^s} v^{(n),j} v^{(n),t} \tag{34}$$

$$\frac{\partial^2 f}{\partial u_j^i \partial w_t^{(k),s}} = \frac{\partial f}{\partial p^i} \frac{\partial v^{(n),j}}{\partial w_t^{(k),s}} + \frac{\partial^2 f}{\partial p^i \partial p^l} v^{(n),j} \frac{\partial p^l}{\partial v^{(n),m}} \frac{\partial v^{(n),m}}{\partial w_t^{(k),s}}$$

$$= \frac{\partial f}{\partial p^i} \eta_s^{(n,k),j} v^{(k-1),t} + \frac{\partial^2 f}{\partial p^i \partial p^l} v^{(n),j} u_m^l \eta_s^{(n,k),m} v^{(k-1),t} \tag{35}$$

$$\frac{\partial^2 f}{\partial w_j^{(k),i} \partial w_t^{(r),s}} = \frac{\partial^2 f}{\partial p^l \partial p^q} \frac{\partial p^l}{\partial v^{(n),m}} \frac{\partial v^{(n),m}}{\partial w_j^{(k),i}} \frac{\partial p^q}{\partial v^{(n),a}} \frac{\partial v^{(n),a}}{\partial w_t^{(r),s}} + \frac{\partial f}{\partial p^l} \frac{\partial p^l}{\partial v^{(n),m}} \frac{\partial^2 v^{(n),m}}{\partial w_j^{(k),i} \partial w_t^{(r),s}}$$

$$= \frac{\partial^2 f}{\partial p^l \partial p^q} u_m^l \eta_i^{(n,k),m} v^{(k-1),j} u_a^q \eta_s^{(n,r),a} v^{(r-1),t}$$

$$+ \frac{\partial f}{\partial p^l} u_m^l \times \begin{cases} \eta_s^{(n,r),m} \eta_i^{(r-1,k),t} v^{(k-1),j} & r > k \\ 0 & r = k \\ \eta_i^{(n,k),m} \eta_s^{(k-1,r),j} v^{(r-1),t} & r < k \end{cases} \tag{36}$$

To use CR, we calculate $\alpha$ as

$$\alpha = \frac{-\mathbf{g}^T \mathbf{H} \mathbf{g} + \sqrt{\left(\mathbf{g}^T \mathbf{H} \mathbf{g}\right)^2 - 2\left(\mathbf{g}^T \nabla f\right)\left(\sigma \|\mathbf{g}\|^3\right)}}{\sigma \|\mathbf{g}\|^3} \tag{37}$$

For a given iteration for the deep network described above (dropping the subscript $j$'s so as not to interfere with the index notation), the quantities in this equation are

$$\mathbf{g}^T \nabla f = \frac{\partial F}{\partial p^i} v^{(n),j} \omega_j^i + \sum_k \frac{\partial F}{\partial p^l} u_m^l \eta_i^{(n,k),m} v^{(k-1),j} \phi_j^{(k),i} \tag{38}$$

$$\mathbf{g}^T \mathbf{H} \mathbf{g} = \omega_j^i \frac{\partial^2 F}{\partial p^i \partial p^s} v^{(n),j} v^{(n),t} \omega_t^s$$

$$+ 2 \sum_k \omega_j^i \left( \frac{\partial F}{\partial p^i} \eta_s^{(n,k),j} v^{(k-1),t} + \frac{\partial^2 F}{\partial p^i \partial p^l} v^{(n),j} u_m^l \eta_s^{(n,k),m} v^{(k-1),t} \right) \phi_t^{(k),s}$$

$$+ \sum_k \phi_j^{(k),i} \frac{\partial^2 F}{\partial p^l \partial p^q} u_m^l \eta_i^{(n,k),m} v^{(k-1),j} u_a^q \eta_s^{(n,k),a} v^{(k-1),t} \phi_t^{(k),s}$$

$$+ 2 \sum_{k=2}^n \sum_{r=1}^{k-1} \phi_j^{(k),i} \frac{\partial F}{\partial p^l} u_m^l \eta_i^{(n,k),m} \eta_s^{(k-1,r),j} v^{(r-1),t} \phi_t^{(r),s} \tag{39}$$

$$\|\mathbf{g}\|^3 = \left( \omega_j^i \omega_j^i + \sum_k \phi_j^{(k),i} \phi_j^{(k),i} \right)^{\frac{3}{2}} \tag{40}$$

With these formulae in hand, and using the generalized binomial theorem ($\sqrt{1+\epsilon} \approx 1 + \epsilon/2$ for $\epsilon \ll 1$), we can consider what happens to $\alpha$ in limiting cases:

$$\alpha \approx \begin{cases} \frac{|\mathbf{g}^T \nabla f|}{\mathbf{g}^T \mathbf{H} \mathbf{g}} & |\mathbf{g}^T \mathbf{H} \mathbf{g}| \gg |\mathbf{g}^T \nabla f| \left( \sigma \|\mathbf{g}\|^3 \right) \text{ and } \mathbf{g}^T \mathbf{H} \mathbf{g} > 0 \\ \sqrt{\frac{2\mathbf{g}^T \nabla f}{\sigma \|\mathbf{g}\|^3}} & |\mathbf{g}^T \mathbf{H} \mathbf{g}| \ll |\mathbf{g}^T \nabla f| \left( \sigma \|\mathbf{g}\|^3 \right) \text{ and } \mathbf{g}^T \mathbf{H} \mathbf{g} > 0 \\ 2 \left| \frac{\mathbf{g}^T \mathbf{H} \mathbf{g}}{\sigma \|\mathbf{g}\|^3} \right| + \left| \frac{\mathbf{g}^T \nabla f}{\mathbf{g}^T \mathbf{H} \mathbf{g}} \right| & |\mathbf{g}^T \mathbf{H} \mathbf{g}| \gg |\mathbf{g}^T \nabla f| \left( \sigma \|\mathbf{g}\|^3 \right) \text{ and } \mathbf{g}^T \mathbf{H} \mathbf{g} < 0 \end{cases} \tag{41}$$

The weight update scheme for a single learning rate at each iteration is

$$\mathbf{u}_{j+1} = \mathbf{u}_j + \alpha_j \boldsymbol{\omega}_j \tag{42}$$

$$\mathbf{w}_{j+1}^{(k)} = \mathbf{w}_j^{(k)} + \alpha_j \boldsymbol{\phi}_j^{(k)} \tag{43}$$

We could instead consider a weight-specific learning rate. If we assume that the baseline $\mathbf{g}$ vector for each variable is a unit step in the direction of that variable, and we ignore superscripts indicating the iteration number, the calculations for variable-specific learning rates $\alpha_j^{(u),i}$ for $u_j^i$ and $\alpha_j^{(w,k),i}$ are as shown in Table 2:

Table 2: Variable-Specific Learning Rate Calculations

| Learning Rate | $\mathbf{g}^t \nabla f$ | $\mathbf{g}^T \mathbf{H} \mathbf{g}$ | $\|\mathbf{g}\|^3$ |
|---|---|---|---|
| $\alpha_j^{(u),i}$ | $\frac{\partial F}{\partial p^i} v^{(n),j}$ | $\frac{\partial^2 F}{\partial p^i \partial p^s} v^{(n),j} v^{(n),t}$ | 1 |
| $\alpha_j^{(w,k),i}$ | $\frac{\partial F}{\partial p^l} u_m^l \eta_i^{(n,k),m} v^{(k-1),j}$ | $\frac{\partial^2 F}{\partial p^l \partial p^q} u_m^l \eta_i^{(n,k),m} v^{(k-1),j} u_a^q \eta_s^{(n,k),a} v^{(k-1),t}$ | 1 |

## B.2 CONVOLUTIONAL AND RECURRENT LAYERS

Convolutional and recurrent layers preserve the low-rank derivative structure of the fully connected feedforward layers considered above, and we will show this in the following sections. Because we are only considering a single layer of each, we calculate the derivatives of the layer outputs with respect to the layer inputs – in a larger network, those derivatives will be necessary for calculating total derivatives via back-propagation.

### B.2.1 Convolutional Layer

We can define a convolutional layer as

$$v_t^s = \mathcal{A}\left(\left(\tilde{x}_t^s\right)_k^l w_k^l\right) \tag{44}$$

$$\left(\tilde{x}_t^s\right)_k^l = x_{\tau t+k-1}^{\sigma s+l-1} \tag{45}$$

where $x_j^i$ is the layer input, $\sigma$ is the vertical stride, $\tau$ is the horizontal stride, $\mathcal{A}$ is the activation function, and $v_t^s$ is the layer output. A convolutional structure can make the expressions somewhat complicated when expressed in index notation, but we can simplify matters by using the simplification $z_{tk}^{sl} = x_{\tau t+k-1}^{\sigma s+l-1}$. The layer definition is then

$$v_t^s = \mathcal{A}\left(z_{tk}^{sl} w_k^l\right) \tag{46}$$

The derivatives of the convolutional layer are

$$\frac{\partial v_t^s}{\partial x_j^i} = \mathcal{A}'\left(z_{tk}^{sl} w_k^l\right) w_p^m \frac{\partial z_{tq}^{sm}}{\partial x_j^i} \tag{47}$$

$$\frac{\partial z_{tq}^{sm}}{\partial x_j^i} = \begin{cases} 1 & i = \sigma s + m - 1 \text{ and } j = \tau t + p - 1 \\ 0 & \text{else} \end{cases} \tag{48}$$

$$\frac{\partial v_t^s}{\partial w_q^p} = \mathcal{A}'\left(z_{tk}^{sl} w_k^l\right) z_{tq}^{sp} \tag{49}$$

$$\frac{\partial^2 v_t^s}{\partial x_q^p \partial x_j^i} = \mathcal{A}''\left(z_{tk}^{sl} w_k^l\right) w_r^m \frac{\partial z_{tr}^{sm}}{\partial x_j^i} w_b^a \frac{\partial z_{tb}^{sa}}{\partial x_q^p} \tag{50}$$

$$\frac{\partial^2 v_t^s}{\partial w_q^p \partial x_j^i} = \mathcal{A}'\left(z_{tk}^{sl} w_k^l\right) \frac{\partial z_{tq}^{sp}}{\partial x_j^i} + \mathcal{A}''\left(z_{tk}^{sl} w_k^l\right) z_{tq}^{sp} w_r^m \frac{\partial z_{tr}^{sm}}{\partial x_j^i} \tag{51}$$

$$\frac{\partial^2 v_t^s}{\partial w_q^p \partial w_b^a} = \mathcal{A}''\left(z_{tk}^{sl} w_k^l\right) z_{tq}^{sp} z_{tb}^{sa} \tag{52}$$

with no summation over $s$ and $t$ in any of the expressions above. Using the simplification with $z_{tk}^{sl}$ makes it significantly easier to see the low rank structure in these derivatives, but that structure is still noticeable without the simplification.

$$\frac{\partial v_t^s}{\partial w_q^p} = \mathcal{A}'\left(\left(\tilde{x}_t^s\right)_k^l w_k^l\right)\left(\tilde{x}_t^s\right)_q^p = \mathcal{A}'\left(\left(\tilde{x}_t^s\right)_k^l w_k^l\right) x_{\tau t+q-1}^{\sigma s+p-1} \tag{53}$$

$$\frac{\partial v_t^s}{\partial x_j^i} = \begin{cases} \mathcal{A}'\left(\left(\tilde{x}_t^s\right)_k^l w_k^l\right) w_{j+1-\tau t}^{i+1-\sigma s} & i+1 > \sigma s, j+1 > \tau t \\ 0 & \text{else} \end{cases} \tag{54}$$

$$\frac{\partial^2 v_t^s}{\partial w_q^p \partial w_b^a} = \mathcal{A}''\left(\left(\tilde{x}_t^s\right)_k^l w_k^l\right) x_{\tau t+q-1}^{\sigma s+p-1} x_{\tau t+b-1}^{\sigma s+a-1} \tag{55}$$

$$\frac{\partial^2 v_t^s}{\partial w_q^p \partial x_j^i}$$

$$= \begin{cases} \mathcal{A}'\left(\left(\tilde{x}_t^s\right)_k^l w_k^l\right) + \mathcal{A}''\left(\left(\tilde{x}_t^s\right)_k^l w_k^l\right) w_{j+1-\tau t}^{i+1-\sigma s} x_{\tau t+q-1}^{\sigma s+p-1} & i+1-\sigma s = p, j+1-\tau t = q \\[2mm] \mathcal{A}''\left(\left(\tilde{x}_t^s\right)_k^l w_k^l\right) w_{j+1-\tau t}^{i+1-\sigma s} x_{\tau t+q-1}^{\sigma s+p-1} & \begin{aligned} i+1-\sigma s &> 0, i+1-\sigma s \neq p, \\ j+1-\tau t &> 0, j+1-\tau t \neq q \end{aligned} \\[2mm] 0 & \text{else} \end{cases} \tag{56}$$

$$\frac{\partial^2 v_t^s}{\partial x_q^p \partial x_j^i} = \begin{cases} \mathcal{A}''\left(\left(\tilde{x}_t^s\right)_k^l w_k^l\right) w_{j+1-\tau t}^{i+1-\sigma s} w_{q+1-\tau t}^{p+1-\sigma s} & \begin{aligned} p+1 &> \sigma s, i+1 > \sigma s \\ q+1 &> \tau t, j+1 > \tau t \end{aligned} \\[2mm] 0 & \text{else} \end{cases} \tag{57}$$

The conditional form of the expressions is more complicated, but it is also possible to see how the derivatives relate to $w_j^i$ and submatrices of $x_j^i$.

### B.2.2  RECURSIVE LAYER

We can define our recursive layer as

$$v_{(t)}^j = \mathcal{A}\left(w_i^j v_{(t-1)}^i\right) \tag{58}$$

$$v_{(0)}^j = x^j \tag{59}$$

where $t$ indicates the number of times that the recursion has been looped through. If we inspect this carefully, we can actually see that this is almost identical to the hidden layers of the feedforward network: they are identical if we stipulate that the weights of the feedforward network are identical at each layer (i.e. $w_j^{(k),i} = w_j^i \forall k$) and if we treat the recursive loops like layers. This observation allows us to reuse some of our previous derivations. Primarily, we will use the fact that

$$\frac{\partial\left(\cdot\right)}{\partial w_j^i} = \sum_k \frac{\partial\left(\cdot\right)}{\partial w_p^{(k),m}} \frac{\partial w_j^{(k),i}}{\partial w_j^i} = \sum_k \frac{\partial\left(\cdot\right)}{\partial w_j^{(k),i}} \tag{60}$$

The first-order derivatives are then

$$\frac{\partial v_{(t)}^m}{\partial w_r^s} = \sum_k \eta_s^{(t,k),m} v_{(t-1)}^r \tag{61}$$

$$\frac{\partial v_{(t)}^m}{\partial x^i} = \eta_s^{(t,1),m} w_i^s \tag{62}$$

If $\mathcal{A}$ is a ReLU, then the second derivatives are relatively simple

$$\frac{\partial v_{(t)}^m}{\partial w_j^i \partial w_r^q} = \sum_{k,p} \frac{\partial v_{(t)}^m}{\partial w_j^{(k),i} \partial w_r^{(p),q}}$$

$$= 2 \sum_{k=2}^{t} \sum_{p=1}^{k-1} \eta_i^{(t,k),m} \eta_q^{(k-1,p),j} v_{(p-1)}^r \tag{63}$$

$$\frac{\partial v_{(t)}^m}{\partial x^i \partial x^l} = 0 \tag{64}$$

$$\frac{\partial v_{(t)}^m}{\partial w_r^q \partial x^i} = \eta_s^{(t,1),j} \delta_i^s + \sum_k \frac{\partial \eta_q^{(t,1),m}}{\partial w_t^{(k),s}} w_i^s$$

$$= \eta_s^{(t,1),j} \delta_i^s + \sum_k \eta_q^{(t,k),m} \eta_s^{(k-1,1),r} w_i^s \tag{65}$$

If $\mathcal{A}$ is not a ReLU, then we would use the results in the next section to calculate the second derivatives. Regardless of the exact form of $\mathcal{A}$, though, we retain the low-rank structure as long as $\mathcal{A}$ is an entry-wise function of its arguments.

## B.3  DEEP NETWORK WITH GENERAL ACTIVATION FUNCTIONS

For a deep network with general entry-wise functions, the first derivatives are all identical to the derivations given in Section 2 save that $\mathcal{A}'$ will be different. Before calculating second-order derivatives, though, we do some preliminary calculations that were not necessary before because $\mathcal{A}'' = 0$ for ReLUs. First, we calculate the derivatives of $\gamma_r^{(l),m}$:

$$\frac{\partial \gamma_r^{(l),m}}{\partial w_t^{(q),s}} = \begin{cases} 0 & l < q \\ \delta_r^m \mathcal{A}'' \left( w_j^{(l),m} v^{(l-1),j} \right) \delta_s^m v^{(l-1),t} & l = q \\ \delta_r^m \mathcal{A}'' \left( w_j^{(l),m} v^{(l-1),j} \right) w_a^{(l),m} \eta_s^{(l-1,q),a} v^{(q-1),t} & l > q \end{cases} \tag{66}$$

$$\lambda_{rs}^{(l),m} \equiv \delta_r^m \delta_s^m \mathcal{A}'' \left( w_j^{(l),m} v^{(l-1),j} \right) \tag{67}$$

$$\frac{\partial \gamma_r^{(l),m}}{\partial w_t^{(q),s}} = \begin{cases} 0 & l < q \\ \lambda_{rs}^{(l),m} v^{(l-1),t} & l = q \\ \lambda_{rp}^{(l),m} w_a^{(l),p} \eta_s^{(l-1,q),a} v^{(q-1),t} & l > q \end{cases} \tag{68}$$

where there is no summation over $m$ in any of these equations. Next, we calculate the derivatives of $\alpha_m^{(k,l),j}$:

$$\alpha_m^{(k,l),j} = \alpha_b^{(k,r),j} \alpha_a^{(r,r-1),b} \alpha_m^{(r-1,l),a} \tag{69}$$

$$\alpha_a^{(r,r-1),b} = \beta_a^{(r),b} = \gamma_s^{(r),b} w_a^{(r),s} \tag{70}$$

$$\frac{\partial \beta_a^{(r),b}}{\partial w_t^{(q),s}} = \begin{cases} 0 & r < q \\ \lambda_{ps}^{(r),b} w_a^{(r),p} v^{(r-1),t} + \gamma_s^{(r),b} \delta_a^t & r = q \\ \lambda_{dp}^{(r),b} w_a^{(r),d} w_m^{(r),p} \eta_s^{(r-1,q),m} v^{(q-1),t} & r > q \end{cases} \tag{71}$$

$$\frac{\partial \alpha_m^{(k,l),j}}{\partial w_t^{(q),s}} = \sum_{r=l+1}^{k} \alpha_b^{(k,r),j} \frac{\partial \beta_a^{(r),b}}{\partial w_t^{(q),s}} \alpha_m^{(r-1,l),a}$$

$$= \begin{cases} 0 & k < q \\ \begin{aligned} &\sum_{r=q+1}^{k} \alpha_b^{(k,r),j} \lambda_{dp}^{(r),b} w_a^{(r),d} w_c^{(r),p} \eta_s^{(r-1,q),c} v^{(q-1),t} \alpha_m^{(r-1,l),a} \\ &+ \alpha_b^{(k,q),j} \left( \lambda_{ps}^{(q),b} w_a^{(q),p} v^{(q-1),t} + \gamma_s^{(q),b} \delta_a^t \right) \alpha_m^{(q-1,l),a} \end{aligned} & l < q \le k \\ \sum_{r=l+1}^{k} \alpha_b^{(k,r),j} \lambda_{dp}^{(r),b} w_a^{(r),d} w_c^{(r),p} \eta_s^{(r-1,q),c} v^{(q-1),t} \alpha_m^{(r-1,l),a} & q \le l \end{cases} \tag{72}$$

Thirdly, we calculate the derivatives of $\eta_l^{(n,k),j}$:

$$\frac{\partial \eta_l^{(n,k),j}}{\partial w_t^{(q),s}} = \frac{\partial \alpha_m^{(n,k),j}}{\partial w_t^{(q),s}} \gamma_l^{(k),m} + \alpha_m^{(n,k),j} \frac{\partial \gamma_l^{(k),m}}{\partial w_t^{(q),s}} \tag{73}$$

$$\frac{\partial \eta_l^{(n,k),j}}{\partial w_t^{(q),s}}$$

$$= \begin{cases} \begin{aligned} &\sum_{r=q+1}^{n} \alpha_b^{(n,r),j} \lambda_{dp}^{(r),b} w_a^{(r),d} w_c^{(r),p} \eta_s^{(r-1,q),c} v^{(q-1),t} \alpha_m^{(r-1,k),a} \gamma_l^{(k),m} \\ &+ \alpha_b^{(n,q),j} \left( \lambda_{ps}^{(q),b} w_a^{(q),p} v^{(q-1),t} + \gamma_s^{(q),b} \delta_a^t \right) \alpha_m^{(q-1,k),a} \gamma_l^{(k),m} \end{aligned} & q > k \\ \begin{aligned} &\sum_{r=k+1}^{n} \alpha_b^{(n,r),j} \lambda_{dp}^{(r),b} w_a^{(r),d} w_c^{(r),p} \eta_s^{(r-1,k),c} v^{(k-1),t} \alpha_m^{(r-1,k),a} \gamma_l^{(k),m} \\ &+ \alpha_m^{(n,k),j} \lambda_{ls}^{(k),m} v^{(k-1),t} \end{aligned} & q = k \\ \begin{aligned} &\sum_{r=k+1}^{n} \alpha_b^{(n,r),j} \lambda_{dp}^{(r),b} w_a^{(r),d} w_c^{(r),p} \eta_s^{(r-1,k),c} v^{(k-1),t} \alpha_m^{(r-1,k),a} \gamma_l^{(k),m} \\ &+ \alpha_m^{(n,k),j} \lambda_{lp}^{(k),m} w_a^{(k),p} \eta_s^{(k-1,q),a} v^{(q-1),t} \end{aligned} & q < k \end{cases} \tag{74}$$

$$= \begin{cases} \begin{aligned} &\sum_{r=q+1}^{n} \alpha_b^{(n,r),j} \lambda_{dp}^{(r),b} w_a^{(r),d} w_c^{(r),p} \eta_s^{(r-1,q),c} v^{(q-1),t} \eta_l^{(r-1,k),a} \\ &+ \alpha_b^{(n,q),j} \lambda_{ps}^{(q),b} w_a^{(r),p} v^{(q-1),t} \eta_l^{(q-1,k),a} + \eta_s^{(n,q),j} \eta_l^{(q-1,k),t} \end{aligned} & q > k \\ \begin{aligned} &\sum_{r=k+1}^{n} \alpha_b^{(n,r),j} \lambda_{dp}^{(r),b} w_a^{(r),d} w_c^{(r),p} \eta_s^{(r-1,k),c} v^{(k-1),t} \eta_l^{(r-1,k),a} \\ &+ \alpha_m^{(n,k),j} \lambda_{ls}^{(k),m} v^{(k-1),t} \end{aligned} & q = k \\ \begin{aligned} &\sum_{r=k+1}^{n} \alpha_b^{(n,r),j} \lambda_{dp}^{(r),b} w_a^{(r),d} w_c^{(r),p} \eta_s^{(r-1,k),c} v^{(k-1),t} \eta_l^{(r-1,k),a} \\ &+ \alpha_m^{(n,k),j} \lambda_{lp}^{(k),m} w_a^{(k),p} \eta_s^{(k-1,q),a} v^{(q-1),t} \end{aligned} & q < k \end{cases} \tag{75}$$

$$\frac{\partial^2 v^{(n),j}}{\partial w_i^{(k),l} \partial w_t^{(q),s}} = \begin{cases} \frac{\partial \eta_l^{(n,k),j}}{\partial w_t^{(q),s}} v^{(k-1),i} & q \geq k \\ \frac{\partial \eta_l^{(n,k),j}}{\partial w_t^{(q),s}} v^{(k-1),i} + \eta_l^{(n,k),j} \frac{\partial v^{(k-1),i}}{\partial w_t^{(q),s}} & q < k \end{cases} \tag{76}$$

$$= \begin{cases} \sum_{r=q+1}^{n} \alpha_b^{(n,r),j} \lambda_{dp}^{(r),b} w_a^{(r),d} w_c^{(r),p} \eta_s^{(r-1,q),c} v^{(q-1),t} \eta_l^{(r-1,k),a} v^{(k-1),i} \\ + \alpha_b^{(n,q),j} \lambda_{ps}^{(q),b} w_a^{(q),p} v^{(q-1),t} \eta_l^{(q-1,k),a} v^{(k-1),i} + \eta_s^{(n,q),j} \eta_l^{(q-1,k),t} v^{(k-1),i} & q > k \\[2ex] \sum_{r=k+1}^{n} \alpha_b^{(n,r),j} \lambda_{dp}^{(r),b} w_a^{(r),d} w_c^{(r),p} \eta_s^{(r-1,k),c} v^{(k-1),t} \eta_l^{(r-1,k),a} v^{(k-1),i} \\ + \alpha_m^{(n,k),j} \lambda_{ls}^{(k),m} v^{(k-1),t} v^{(k-1),i} & q = k \\[2ex] \sum_{r=k+1}^{n} \alpha_b^{(n,r),j} \lambda_{dp}^{(r),b} w_a^{(r),d} w_c^{(r),p} \eta_s^{(r-1,k),c} v^{(k-1),t} \eta_l^{(r-1,k),a} v^{(k-1),i} \\ + \alpha_m^{(n,k),j} \lambda_{lp}^{(k),m} w_a^{(k),p} \eta_s^{(k-1,q),a} v^{(q-1),t} v^{(k-1),i} + \eta_l^{(n,k),j} \eta_s^{(k-1,q),i} v^{(q-1),t} & q < k \end{cases} \tag{77}$$

The second-order derivatives of the objective function are then

$$\frac{\partial^2 F}{\partial u_j^i \partial u_t^s} = \frac{\partial^2 F}{\partial p^k \partial p^l} \frac{\partial p^k}{\partial u_j^i} \frac{\partial p^l}{\partial u_t^s} = \frac{\partial^2 F}{\partial p^i \partial p^s} v^{(n),j} v^{(n),t} \tag{78}$$

$$\frac{\partial^2 F}{\partial u_j^i \partial w_t^{(k),s}} = \frac{\partial F}{\partial p^i} \frac{\partial v^{(n),j}}{\partial w_t^{(k),s}} + \frac{\partial^2 F}{\partial p^i \partial p^l} v^{(n),j} \frac{\partial p^l}{\partial v^{(n),m}} \frac{\partial v^{(n),m}}{\partial w_t^{(k),s}}$$
$$= \frac{\partial F}{\partial p^i} \eta_s^{(n,k),j} v^{(k-1),t} + \frac{\partial^2 F}{\partial p^i \partial p^l} v^{(n),j} u_m^l \eta_s^{(n,k),m} v^{(k-1),t} \tag{79}$$

$$\frac{\partial^2 F}{\partial w_i^{(k),l} \partial w_t^{(q),s}} = \frac{\partial^2 F}{\partial p^a \partial p^b} \frac{\partial p^a}{\partial v^{(n),j}} \frac{\partial v^{(n),j}}{\partial w_i^{(k),l}} \frac{\partial p^b}{\partial v^{(n),c}} \frac{\partial v^{(n),c}}{\partial w_t^{(q),s}} + \frac{\partial F}{\partial p^m} \frac{\partial p^m}{\partial v^{(n),j}} \frac{\partial^2 v^{(n),j}}{\partial w_i^{(k),l} \partial w_t^{(q),s}}$$
$$= \frac{\partial^2 F}{\partial p^a \partial p^b} u_j^a \eta_l^{(n,k),j} v^{(k-1),i} u_c^b \eta_s^{(n,q),c} v^{(q-1),t} + \frac{\partial F}{\partial p^a} u_j^a \frac{\partial^2 v^{(n),j}}{\partial w_i^{(k),l} \partial w_t^{(q),s}} \tag{80}$$

