# OpenReview forum: "Understanding and Exploiting the Low-Rank Structure of Deep Networks"
_ICLR.cc/2018/Conference — Reject_

### Official Review · AnonReviewer2 · 2017-11-24
**Neither theoretical nor extensive empirical evidence that the method works is provided**

**Rating:** 4
**Confidence:** 4

**Review:**

[Main comments]

* The authors made a really odd choice of notation, which made the equations hard to follow.
Apparently, that notation is used in differential geometry, but I have never seen it used in
an ML paper. If you talk about outer product structure, show some outer products!

* The function f that the authors differentiate is not even defined in the main manuscript!

* The low-rank structure they describe only holds for a single sample at a time.
I don't see how this would be "understanding low rank structure of deep networks"
as the title claims... What is described is basically an implementation trick.

* Introducing cubic regularization seems interesting. However, either some
extensive empirical evidence or some some theoretical evidence that this is useful are needed.
The present paper has neither (the empirical evidence shown is very limited).

[Other minor comments]

* Strictly speaking Adagrad has not been designed for Deep Learning.
It is an online algorithm that became popular in the DL community later on.

* "Second derivatives should suffice for now, but of course if a use arose for
third derivatives, calculating them would be a real option"

That sentence seems useless.

* Missing citation:

Gradient Descent Efficiently Finds the Cubic-Regularized Non-Convex Newton Step.
Yair Carmon, John Duchi.

---

### Official Review · AnonReviewer1 · 2017-11-25
**Understanding and Exploiting the Low-Rank Structure of Deep Networks**

**Rating:** 5
**Confidence:** 4

**Review:**

Summary:
This paper shows the feedforward network (with ReLU activation functions in the hidden layers, softmax at the output, and cross entropy-loss) exhibits a low-rank derivative structure, which is able to use second-order information without approximating Hessian. For numerical experiments, the author(s) implemented Cubic Regularization on this network structure with SGD (on MNIST and CIFAR10) and Adagrad and Adadelta (on MNIST).

Comments:
The idea of showing low rank structure which makes it possible to use second-order information without approximations is interesting. This feedforward network with ReLU activation, output softmax and cross-entropy-loss is well-known structure for neural networks.

I have some comments and questions as follows.

Have you tried to apply this to another architecture of neural networks? Do you think whether your approach is able to apply to convolutional neural networks, which are widely used?

There is no gain on using CR with Adam as you mention in Discussion part of the paper. Do you think that CR with SGD (or with Adagrad and Adadelta) can be better than Adam? If not, why do people should consider this approach, which is more complicated, since Adam is widely used?

The author(s) should do more experiments to various dataset to be more convincing.

I do like the idea of the paper, but at the current state, it is hard to evaluate the effective of this paper. I hope the author(s) could provide more experiments on different datasets. I would suggest to also try SVHN or CIFAR100. And if possible, please also consider CNN even if you are not able to provide any theory.

---

### Official Review · AnonReviewer3 · 2017-12-02
**Paper based on potentially useful ideas that has a long way to go**

**Rating:** 2
**Confidence:** 4

**Review:**


This paper proposes to set a global step size gradient-based optimization algorithms such as SGD and Adam using second order information. Instead of using second-order information to compute the update directly (as is done in e.g. Newton method), it is used to estimate the change of the objective function in a pre-computed direction. This is computationally much cheaper than full Newton because (a) the Hessian does not need to be inverted (b) vector-Hessian multiplication is only O(#parameters) for a single sample.

There are many issues.

### runtime and computational issues ###

Firstly, the paper does not clearly specify the algorithm it espouses. It states: "once the step direction had been determined, we considered that fixed, took the average of gT Hg and gT ∇f over all of the sample points to produce m (α) and then solved for a single αj value" You should present pseudo-code for this computation and not leave the reader to determine the detailed order of computation for himself. As it stands, it is not only difficult for the reader to infer these details, but also laborious to determine the computational cost per iteration on some network the reader might wish to apply your algorithm to. Since the paper discusses the computational cost of CR only in vague terms, you should at least provide pseudo-code.

Specifically, consider equation (80) at the very end of the appendix and consider the very last term in that equation. It contains d^2v/dwdw. This is a "heavy" term containing the second derivative of the last hidden layer with respect to weights. You do not specify how you compute this term or quantities involving this term. In a ReLU network, this term is zero due to local linearity, but since you claim that your algorithm is applicable to general networks, this term needs to be analyzed further.

While the precise algorithm you suggest is unclear, it's purpose is also unclear. You only use the Hessian to compute the g^THg terms, i.e. for Hessian-vector multiplication. But it is well-known that Hessian-vector multiplication is "relatively cheap" in deep networks and this fact has been used for several algorithms, e.g. http://www.iro.umontreal.ca/~lisa/pointeurs/ECML2011_CAE.pdf and https://arxiv.org/pdf/1706.04859.pdf. How is your method for computing g^THg different and why is it superior?

Also note that the low-rank structure of deep gradients is well-known and not a contribution of this paper. See e.g. https://www.usenix.org/system/files/conference/atc17/atc17-zhang.pdf

### Experiments ###

The experiments are very weak. In a network where weights are initialized to sensible values, your algorithm is shown not to improve upon straight SGD. You only demonstrate superior results when the weights are badly initialized. However, there are a very large number of techniques already that avoid the "SGD on ReLU network with bad initial weights" problem. The most well-known are batch normalization, He initialization and Adam but there are many others. I don't think it's a stretch to consider that problem "solved". Your algorithm is not shown to address any other problems, but what's worse is that it doesn't even seem to address that problem well. While your learning curves are better than straight SGD, I suspect they are well below the respective curves for He init or batchnorm. In any case, you would need to compare your algorithm against these state-of-the-art methods if your goal is to overcome bad initializations. Also, in appendix A, you state that CR can't even address weights that were initialized to values that are too large.

You claim that your algorithm helps with "overcoming plateaus". While I have heard the claim that deep network optimization suffers from intermediate plateaus before, I have not seen a paper studying / demonstrating this behavior. I suggest you cite several papers that do this and then replicate the plateau situations that arose in those papers and show that CR overcomes them, instead of resorting to a platenau situation that is essentially artificially induced by intentionally bad hyperparameter choices.

I do not understand why your initial learning rate for SGD in figures 2 and 3 (0.02 and 0.01 respectively) differ so much from the initial learning rate under CR. Aren't you trying to show that CR can find the "correct" learning rate? Wouldn't that suggest that initial learning rate for SGD should be comparable to the early learning rates chosen by CR? Wouldn't that suggest you should start SGD with a learning rate of around 2 and 0.35 respectively? Since you are annealing the learning rate for SGD, it's going to decline and get close to 0.02 / 0.01 anyway at some point. While this may not be as good as CR or indeed batchnorm or Adam, the blue constant curve you are showing does not seem to be a fair representation of what SGD can do.

You say the minibatch size is 32. For MNIST, this means that 1 epoch is around 1500 iterations. That means your plots only show the first epoch of training. But MNIST does not converge in 1 epoch. You should show the error curve until convergence is reached. Same for CIFAR.

"we are not interested in network performance measures such as accuracy and validation error" I strongly suspect your readers may be interested in those things. You should show validation classification error or at least training classification error in addition to cross-entropy error.

"we will also focus on optimization iteration rather than wall clock time" Again, your readers care more about the latter. You need to show either error curves by clock time or the total time to convergence or supplement your iteration-based graphs with a detailed discussion of how long an iteration takes.

The scope of the experiments is limited because only a single network architecture is considered, and it is not a state-of-the art architecture (no convolution, no normalization mechanism, no skip connections).

You state that you ran experiments on Adam, Adadelta and Adagrad, but you do not show the Adam results. You say in the text that they were the least favorable for CR. This suggests that you omitted the detailed results because they were unfavorable to you. This is, of course, unacceptable!

### (Un)suitability of ReLU for second-order analysis ###

You claim to use second-order information over the network to set the step size. Unfortuantely, ReLU networks do not have second-order information! They are locally linear. All their nonlinearity is contained in non-differentiable region boundaries. While this may lead to the Hessian being cheaper to compute, it means it is not representative of the actual behavior of the network. In fact, the only second-order information that is brought to bear in your experiments is the second-order information of the error function. I am not saying that this particular second-order information could not be useful, but you need to make a distinction in your paper between network second-order info and error function second-order info and make explicit that you only use the former in your experiments. As far as I know, most second-order papers use either tanh or a smoothed ReLU (such as the smoothed hinge used recently by Koh & Liang (https://arxiv.org/pdf/1703.04730.pdf)) for experiments to overcome the local linearity.

### The \sigma hyperparameter ###

You claim that \sigma is not as important / hard to set as \alpha in SGD or Adam. You state: "We also found that this ap- proach requires less problem-specific information (e.g. an optimal initial learning rate) than other first-order methods in order to perform well." You have not provided sufficient evidence for this claim. You say that \sigma can be chosen by considering powers of 10. In many networks, choosing \alpha by considering powers of 10 is sufficient! Even if powers of 2 are considered for \alpha, this would reduce the search effort only by factor log_2(10). Also, what if the range of \sigma values that need to be considered is larger than the range of \alpha values? Then setting \sigma would take more effort.

You do not give precise protocols how you set \sigma and how you set \alpha for non-CR algorithms. This should be clearly specified in Appendix A as it is central to your argument of easing hyperparameter search.

### Minor points ###

- Your introduction could benefit from a few more citations
- "The rank of the weighted sum of low rank components (as occurs with mini-batch sampling) is generally larger than the rank of the summed components, however." I don't understand this. Every sum can be viewed as a weighted sum and vice versa.
- Equation (8) could be motivated a bit better. I know it derives from Taylor's theorem, but it might be good to discuss how Taylor's theorem (and its assumptions) relate to deep networks.
- why the name "cubic regularization"? shouldn't it be something like "quadratic step size tuning"?

.
.
.

The reason I am giving a 2 instead of a 1 is because the core idea behind the algorithm given seems to me to have potential, but the execution is sorely lacking.

A final suggestion: You advertise as one of your algorithms upsides that it uses exact Hessian information. Howwever, since you only care about the scale of the second-order term and not its direction, I suspect exact calculation is far from necessary and you could get away with very cheap approximations, using for example techniques such as mean field analysis (e.g. http://papers.nips.cc/paper/6322-exponential-expressivity-in-deep-neural-networks-through-transient-chaos.pdf).

---

### Decision · Program_Chairs · 2018-01-29
**ICLR 2018 Conference Acceptance Decision**

**Decision:**

Reject

**Comment:**

The reviewers thought that idea of trying to exploit low-rank structure in the loss gradients of a feedforward network to improve training was interesting; however they expressed many concerns about the clarity of the presentation, quality of the empirical evaluation, and significance of the result (since the tests were not done on an architecture anywhere near state-of-the-art). Because the authors did not participate in the discussion period, none of these concerns were addressed.

Pros:
+ Promising idea for new approaches to optimization.

Cons:
- Unclear notation for the intended machine learning audience
- Algorithm should be illustrated using pseudocode
- Limited significance if the method is only usable with purely feedforward networks.
- Limited empirical evaluation: positive results only if weights are poorly initialized.